# Shortcut Approaches to Substance Delivery into the Brain Based on Intranasal Administration Using Nanodelivery Strategies for Insulin

**DOI:** 10.3390/molecules25215188

**Published:** 2020-11-07

**Authors:** Toshihiko Tashima

**Affiliations:** Tashima Laboratories of Arts and Sciences, 1239-5 Toriyama-cho, Kohoku-ku, Yokohama, Kanagawa 222-0035, Japan; tashima_lab@yahoo.co.jp

**Keywords:** intranasal drug administration, nose-to-brain drug delivery, drug delivery system, insulin delivery to brain, olfactory nerve route, nanodelivery

## Abstract

The direct delivery of central nervous system (CNS) drugs into the brain after administration is an ideal concept due to its effectiveness and non-toxicity. However, the blood–brain barrier (BBB) prevents drugs from penetrating the capillary endothelial cells, blocking their entry into the brain. Thus, alternative approaches must be developed. The nasal cavity directly leads from the olfactory epithelium to the brain through the cribriform plate of the skull bone. Nose-to-brain drug delivery could solve the BBB-related repulsion problem. Recently, it has been revealed that insulin improved Alzheimer’s disease (AD)-related dementia. Several ongoing AD clinical trials investigate the use of intranasal insulin delivery. Related to the real trajectory, intranasal labeled-insulins demonstrated distribution into the brain not only along the olfactory nerve but also the trigeminal nerve. Nonetheless, intranasally administered insulin was delivered into the brain. Therefore, insulin conjugates with covalent or non-covalent cargos, such as AD or other CNS drugs, could potentially contribute to a promising strategy to cure CNS-related diseases. In this review, I will introduce the CNS drug delivery approach into the brain using nanodelivery strategies for insulin through transcellular routes based on receptor-mediated transcytosis or through paracellular routes based on escaping the tight junction at the olfactory epithelium.

## 1. Introduction

The blood–brain barrier (BBB) pharmacokinetically blocks the entry of therapeutic agents into the brain. This impermeability represents a serious problem in drug discovery or the development of efficient treatments for central nervous system (CNS)-related diseases such as Alzheimer’s disease (AD) and Parkinson’s disease (PD) [1]. Therefore, it is essential to develop approaches that enable substances to cross directly or indirectly the BBB for better brain-targeted drug delivery. The BBB is virtually composed of (i) a physical barrier based on the tight junctions of the capillary endothelial cells supported by adjacent glial cells and (ii) excreted components from endothelial cells by ATP-binding cassette (ABC) transporters such as multiple drug resistance 1 (MDR1, P-glycoprotein). However, the brain needs nutritional materials such as glucose and has to be able to eliminate waste materials. Solute carrier (SLC) transporters [2] expressed at the BBB allow the entry of low-molecular-weight hydrophilic nutrients. Furthermore, receptor-mediated transcytosis (RMT) distributes the corresponding middle- or high-molecular-weight ligands, such as insulin and transferrin, across the capillary endothelium to the brain. Therefore, carrier-mediated transport using arbitrary substrates with cargos could be a direct approach to bypass MDR1 and the tight junction-based physical boundary at the BBB [3]. Similarly, RMT using arbitrary ligands with cargos could be a direct approach to bypass tight junction-based physical boundaries at the BBB [4]. In general, substances could be classified as low-molecular-weight compounds (<approximately 500 Da), high-molecular-weight compounds (>approximately 3000 Da), and middle-molecular-weight compounds (approximately 500–3000 Da). Drug delivery strategies depend on features such as molecular size, physicochemical properties, such as hydrophobicity or hydrophilicity, enzymatic stability, and affinity to intermedium molecules, such as transporters and receptors, or target molecules. It is true that direct methods to cross the BBB using carrier-mediated transport and RMT could indeed be promising solutions, but the off-target side effects should also be considered that might occur due to incorrect molecular collisions through the systemic circulation, although a certain level of tissue-specific SLC transporter and receptor expression could be observed. Thus, an alternative bypass approach for a brain-targeted substance delivery should be developed. It is well-known that intranasal molecules enter the brain. Accordingly, intranasally administered substances could be delivered into the brain through a nose-to-brain route [5,6].

Drug delivery across the cell membrane attracts significant attention in various biomedical fields, including medicinal chemistry. Furthermore, nanoparticles based on nanodelivery strategies have been extraordinarily used for membrane permeation. I have already described carrier-mediated transport methods based on tissue-specifically expressed transporters using arbitrary substrate–cargo conjugates [3], molecular cross-membrane internalization into the cells using cell-penetrating peptides (CPPs) [7], specific delivery into cancer cells based on receptor-mediated endocytosis [8], and RMT-based brain-targeted delivery across the BBB using arbitrary ligand–cargo conjugates [4]. A constantly growing number of new results allows us to gain a broader view and a more in-depth understanding of the cross-membrane drug delivery. In this perspective review, I introduce a shortcut approach to deliver pharmaceutical agents into the brain through intranasal administration, particularly focusing on insulin, with or without nanoparticles using nano drug delivery systems, as shown in Figure 1.

## 2. Discussions

### 2.1. Anatomical Features of the Nasal Cavity

The proper development of the olfactory system is essential for animals, particularly nocturnal and forest animals, to survive difficult circumstances by quickly smelling scents in the air. Humans could feel spring in the air due to the volatile organic compounds emitted by plants. Vapored molecules of geraniol, a scent ingredient in roses, drift into the nasal cavity through the nostrils, and they are caught by nasal mucosa at the olfactory epithelium. Subsequently, they bind to the seven transmembrane G protein-coupled odorant receptors expressed in non-motile cilia on the dendrites of bipolar olfactory sensory neurons of the olfactory nerve (CN I). These bindings trigger neuronal activation, and the resulting odor signal impulses are electrically transmitted to the glomeruli along the olfactory sensory neuronal axons that penetrate through the cribriform plate of the skull bone into the brain. Within the glomeruli in the olfactory bulb, the information is transmitted through synaptic connectivity, using glutamates as major neurotransmitters [9], to mitral and tufted cells that function as secondary olfactory neurons. Their axons stretch in the olfactory tract of the olfactory bulb and electrically transmit the impulse to the olfactory cortex and higher olfactory center. As a result, chemical stimulation is perceived as a delicate and blissful scent in the brain. Exogenous smell signal transduction occurs from the nose to the brain, which is consistent with scientific materialism. The human nasal cavity (area of approximately 150–160 cm^2^) could be divided into three regions: the vestibular (approximately 0.6 cm^2^), olfactory (approximately 10 cm^2^), and respiratory (approximately 130 cm^2^) regions. In the olfactory region, the olfactory epithelium, covered with mucosa, is composed of olfactory sensory neurons, supporting cells, basal cells, and Bowman’s glands that secrete mucin, including sialic acid and sulfate. The lamina propria between the olfactory epithelium and cribriform plate contains blood vessels and lymphatic vessels (Figure 1 and Figure 2) [6,10]. This cribriform plate-related route is the only anatomic passage that directly connects the outer environment with the brain. This fact has profound significance concerning the brain-targeted drug delivery. However, the trigeminal nerve (CN V), trifurcated into the ophthalmic (V1), the maxillary (V2), and the mandibular (V3) nerves at the trigeminal ganglion, sends signals, such as pain and temperature-sensing, to the CNS. Branches of the ophthalmic nerve innervate the back of the olfactory epithelium, some of which penetrate into the lamina propria through the cribriform plate. Branches of the maxillary nerve innervate the back of the respiratory epithelium. Therefore, the ophthalmic and maxillary nerves also play an important role in the intranasal drug delivery into the brain.

### 2.2. Neurotropic Viruses Entering the Brain Intranasally

Intriguingly, it is known that viruses, such as influenza, enter into the brain transcellularly through or extracellularly along the olfactory nerve that is exposed to the outer environment and is directly connected to the CNS, although the transport mechanisms are not precisely understood. Influenza virus penetration into the brain slightly differed after intranasal inoculation in an in vivo assay using ferrets based on isolation detection or immunohistochemistry, depending on the virus species or strains [11]. Moreover, certain viruses, such as HPAI and H5N1, penetrate the trigeminal nerve in ferrets, in addition to the olfactory nerve [12,13]. The non-myelinated olfactory nerve axon is surrounded by olfactory ensheathing cells that form interstitially conduit open to cerebrospinal fluid (CSF), regardless of olfactory nerve degeneration and regeneration [11]. It is likely that viruses go through such conduits and then escape from there into the CSF. However, viruses penetrate inside the olfactory sensory cells. Viruses have provided hints about drug design for intranasal delivery, such as viral vector plasmid and virus-derived CPPs, to medicinal chemists and pharmaceutical scientists.

### 2.3. Development of Substance Delivery to the Brain through Intranasal Administration

Intranasal administration-mediated drug delivery into the brain attracts considerable attention. This CNS drug administration strategy is an alternative method to cross the BBB and shows good bioavailability due to the considerable transport processes through the olfactory epithelium, rapid distribution to the CNS, and avoidance of first-pass metabolism in the liver [5,6]. Several intranasal drugs, such as oxytocin (1007 Da) and desmopressin (1069 Da), which are administered through the olfactory and respiratory epithelium by spraying, have already been introduced to the market. However, this method is not likely to become a common medication approach compared to the oral and intravenous administrations. Most intranasal drugs that have been developed for systemic or local action, such as anti-allergic rhinitis activity in the nose, are absorbed considerably in the respiratory region and then enter the systemic circulation. Some of these drugs cross the BBB to enter the brain. In contrast, it is likely that intranasal drugs that are absorbed in the olfactory region have not yet been developed for local action in the brain, regardless of the direct connection between the olfactory epithelium and the brain. The difference between the species, e.g., concerning the olfactory space, is one of the problems in drug development. Furthermore, the drug absorption mechanisms through the olfactory epithelium and their successive drug delivery to the brain remain unknown. While the capillary endothelial cells at the BBB contribute to transporting substances to the brain, the olfactory epithelial cells contribute to transmitting olfactory stimulation to the brain as sensors. In other words, olfactory epithelial cells are not good at delivering substances such as nutritive compounds, except for homeostasis maintenance. Thus, the success of intranasal delivery to the brain would rely on ingenious designs and plans.

When intranasally administered drugs are not absorbed through the olfactory epithelium, they move to the respiratory epithelium that forms the nasal cavity, subsequently to the pharynx, and eventually to the stomach full of gastric acid, along with nasal mucosa, due to mucus secretion-based clearance from the submucosal glands and the olfactory cilium stroke that beats appropriate 1000 times every minute [14]. As one of methods to increase the residence time of the drug in the nasal mucosa, it is suggested that mucoadhesive nanocarriers composed of cationic materials electrostatically attach to anionic mucosa and subsequently turn into a gel induced by body temperature there to release gradually the cargos. Chitosan and hydrogels might play such a role. In addition, blood vessels in the respiratory epithelium are significantly vascularized, and drugs absorbed through these cells are thus prone to enter the systemic circulation, although some of them reach, as intended, the CNS along the trigeminal nerve. Therefore, drugs have to be anteromedially and quickly absorbed through the olfactory epithelium. At present, drug delivery using nanoparticle systems has been developed. When nanocarriers are used for nose-to-brain drug delivery to enhance availability, they should properly exhibit protection from degradation, mucoadherance, transparent into mucus layer, cargo release at appropriate sites, and rapid transepithelium of cargo released from or contained in carrier.

### 2.4. Transepithelial Pathways for Substances

#### 2.4.1. Transcellular Pathway

There are two routes for substances to cross the epithelium: the transcellular and the paracellular pathway (Figure 2). The transcellular pathway is a transport process based on crossing both the apical and basolateral membranes via the cytoplasm through passive lipoidal diffusion, transporter-mediated transportation, or endocytosis and subsequent exocytosis-mediated transcytosis. Although hydrophobic low-molecular-weight compounds can penetrate the apical and successively basolateral membrane through passive diffusion or transporter-mediated transportation, middle- and large-molecular-weight compounds cannot easily penetrate the cell membrane. Alternatively, such large substances are transported across the cell through transcytosis [4]. (i) In fact, protein ligands such as insulin, transferrin, and apolipoproteinE (ApoE) could enter the cells through receptor-mediated endocytosis, are contained in endosomes, and then leave the cells through exocytosis on the opposite side. (ii) Monoclonal antibodies also induce endocytosis after binding to the corresponding receptors. (iii) Moreover, CPP binding to the corresponding receptors induces endocytosis, in addition to internalization mechanisms such as direct translocation through the plasma membrane. It is known that TAT (YGRKKRRQRRR as amino acid sequence) as a ligand is responsible for inducing endocytosis after electrostatic binding to anionic heparan sulfate proteoglycans (HSPGs) as the corresponding receptor. If intranasal substance delivery into the brain is carried out based on RMT, receptors that are expressed at the surface of olfactory sensor neurons or olfactory supporting cells at the epithelium might be chosen. The olfactory bulb provides neuronal growth factor (NGF) to immature olfactory sensory nerves. NGF is a ligand of tropomyosin receptor kinase A (TrkA), which is a receptor expressed in immature and mature olfactory sensory nerves in the epithelium, and it is detected by an immunohistochemical study using TrkA (763) antibody [15]. Generally, TrkA demonstrates an intriguing transcytosis trajectory in sympathetic neurons. Recycling endosomes that contained TrkA through constitutive endocytosis were conveyed in the anterograde direction along microtubules by kinesin-1 from the cell body to axonal or dendritic terminals under Rab11 regulation and were fused to the membrane. Signaling endosomes that contained NGF–TrkA through receptor-mediated endocytosis were conveyed in a retrograde direction along microtubules by dynein under Rab5 regulation against early endosomes and Rab7 regulation against late endosomes and modulated gene expression [16,17]. However, the delivery strategy for designed substances such as TrkA ligands to go through two successive transaxonal pathways based on synaptic connectivity between the olfactory sensory nerves and mitral cells might be difficult due to the long distance and poor trafficking control. Nonetheless, after endocytosis, passing from the olfactory sensory nerve to the CSF could be a possible alternative route.

#### 2.4.2. Paracellular Pathway

Another transportation route is the paracellular pathway through the extracellular route between epithelial cells. Epithelial cells form a cell sheet structure-based fluid compartment, control substance transport, and maintain the extracellular environment inside the epithelium. Therefore, epithelial cells tightly adhere to each other at the apicolateral borders to prevent material entry using size- and charge-selective pore filters. Tight junction components between adjacent cell membranes form a such barrier and are composed of transmembrane claudins and occludins, as well as cytosolic zonula occludens (ZO) proteins connecting such transmembrane proteins and actin filaments. In general, the pore diameters enabling paracellular flux range between approximately 0.5 and 1 nm. Permeation enhancers could open tight junction pores in the olfactory epithelium up to approximately 15 nm [18]. Around damaged cells, tight junctions become non-restricted pathways [19]. Olfactory sensory neurons are always exposed to exogenous materials and are subject to injury or cell death due to apoptosis and secondarily regenerated. Stem cells in the basal region undergo differentiation into olfactory sensory neurons or supporting cells. Accordingly, the tight junction at the olfactory epithelium is looser than in other endothelia, such as the BBB and even the small intestinal epithelium. Relatively large molecular substances could pass through the olfactory epithelium through the paracellular pathway. Thus, the paracellular pathway at the olfactory epithelium is an attractive target site for intranasal drug delivery into the brain.

### 2.5. Convective and Bidirectional Flow in the Brain

The current substance transportation network in the brain is suggested to comprise micro drainage such as perineuronal space [20], perivascular space [20,21,22,23,24], and paravascular space [22] and macro drainage such as subarachnoid space for CSF (approximately 140 mL in human adult brain, 450 mL produced per day) and brain parenchyma extracellular space for the interstitial fluid (ISF) (approximately 280 mL in human adult brain), connected to the systemic circulation such as arteries, veins, and the lymphatic system outside of the brain. The lymphatic system plays an important role in transporting waste materials to the venous system. However, it is recognized that the brain does not have a lymphatic system, although waste materials such as amyloid beta (Aβ), tau, and alpha-synuclein (α-syn) must be removed from the brain. It has been revealed that ISF and CSF extracellular perfusion in the brain flush them down the systemic circulation and the lymphatic system through drainage, such as the perivascular and paravascular spaces. This is known as the glymphatic pathway, which is one of the clearance routes in the brain [25,26]. CSF enters from the subarachnoid space into the brain parenchyma through the paravascular space and combines with the ISF. Aquaporin-4 (AQP4) channels on the astrocyte endfeet enhanced ISF flow interstitially along with waste materials through the brain parenchyma from arterial drainage to venous drainage by providing water [27]. During sleep, glymphatic function was enhanced to expel waste materials, which implied the importance of good quality sleep. Intranasal drug availability just before sleep might be reduced as part of the CSF joins the lymphatic system at the olfactory epithelium through the cribriform plate from the brain, which leads to the cervical lymph nodes. Certain other parts of the CSF join the lymphatic system through capillary vessels in the brain and arachnoid granulation. However, the structure and flow direction of the perivascular and paravascular space are controversial, and the nomenclature is likely to be confusing. The paravascular pathway is in the anterograde direction of the bloodstream based on arterial and venous pulsation. The perivascular pathway is retrograde of the bloodstream based on vascular muscle contraction and relaxation. Nonetheless, taking advantage of the convective and bidirectional glymphatic pathways could enable intranasal drug delivery into the brain.

### 2.6. Intranasal Insulin Delivery into the Brain

#### 2.6.1. Transepithelial Insulin Delivery Mechanisms through the Transcellular or Paracellular Pathways

Certain dipeptides and tripeptides are substrates of peptide transporter 1. Certain oligopeptides are transported across the endoplasmic reticulum membrane by translocon [28]. Accordingly, it is turned out that peptide and protein cross-membrane transport is difficult without active transport systems. However, cyclosporine is an *N*-methylated cyclic peptide composed of 11 amino acids (1203 Da) and is produced by non-ribosomal peptide synthetases (Figure 3). Its structure is so compact and hydrophobic that it penetrates cells across the membrane through passive lipoidal diffusion, although cyclosporine is a type of peptide. However, insulin is a relatively large cyclic peptide linked with intramolecular disulfide bonds and is composed of 51 amino acids (5808 Da, A chain; 21 amino acids, B chain; 30 amino acids) (Figure 4 and Figure 5). From the point of such structural features, it is relatively hard, though not impossible, for the insulin to cross the membrane through passive diffusion or direct translocation even together with CPPs such as TAT, due to insulin size and non-*N*-methylated hydrophilicity. It is known that at the BBB, insulin is transported into capillary endothelial cells based on receptor-mediated endocytosis due to insulin receptor (InsR) and subsequently exocytosed to the brain [4]. Endocytosed InsRs are indeed recycled into the plasma membrane, but it is thought that at the surface of the olfactory epithelium, InsRs are probably not expressed at a high level compared to other tissues, as it does not directly lie next to the bloodstream, while InsRs are highly expressed in the olfactory bulb [29]. Notwithstanding, intranasally administered insulin was absorbed in the olfactory epithelium and then reached the brain. Intriguingly, insulin could pass through the capillary endothelium differently than in the case of RMT at the BBB [30]. Alexa Fluor 568-labeled insulin was passed through human adipose microvascular epithelial and aortic endothelial cell monolayers through clathrin-dependent and possibly caveolae-dependent transcytosis, respectively [31]. Therefore, it is suggested that certain types of insulin transportation systems operate through passing the olfactory epithelium barrier. It is noteworthy that some of these transport systems could operate simultaneously, depending on the conditions and/or cell phenotypes. At present, such transport mechanisms remain unknown. Plausible routes are divided into transcellular pathways, including RMT, other types of endocytosis/exocytosis such as macropinocytosis, passive diffusion, direct translocation based on CPPs, and the paracellular pathway.

Insulin itself demonstrated reversible paracellular permeability enhancement by transiently opening tight junctions due to receptor-mediated processes in several in vitro assays using cultured epithelial cell lines such as T84, CaCO2, and HCT-8 [32]. InsR and insulin-like growth factor (IGF) receptors are expressed on the apical membrane and might induce changes in protein synthesis and cytoskeletal structure after insulin binding to them, which probably result in opening tight junctions. This opening is inhibited by cycloheximide [32]. The estimated monomeric insulin diameter is 2.68 nm [33]. Moreover, it was reported that octylglucoside (OG) enhanced insulin permeation across the T-84 monolayer through the paracellular pathway [34]. In addition, permeant inhibitor of phosphatase (PIP) peptide 250 (rrfkvktkkrk -NH_2_) as D-form and PIP peptide 640 (rrdykvevrr-NH_2_) as D-form [35,36] induced myosin light chain phosphorylation to enhance paracellular transport by opening tight junctions. PIP peptide 640 was initially found as an inhibitor of the interaction between protein phosphatase 1 (PP1) and its regulator CPI-17. Insulin co-administered with PIP peptide 250 or 640 reduced blood glucose levels by 50%, respectively, after intraluminal intestinal injection in an in vivo assay using male Wistar rats. Cy3-labeled insulin was detected predominantly in the paracellular route after intraluminal intestinal injection with PIP peptide 640 [36]. Structurally, both PIP peptides 250 and 640 belong to CPPs due to their arginine-rich cationic charge. In fact, they enter cells and subsequently induce phosphorylation to prompt insulin to pass through the paracellular route. So far, the co-internalization of substances with covalent or non-covalent CPPs by endocytosis or direct translocation has often been utilized for drug delivery across the membrane through the transcellular route. Instead, substance delivery through the paracellular route across cells based on tight junction opening induced by cytoplasmic CPPs such as PIP peptides 250 and 640 will play a vital role in drug delivery, although, surprisingly, it was already reported that transactivator of transcription (TAT) caused an alteration of tight junctions to increase paracellular permeability in the retinal pigment epithelium [37]. The intranasal co-administration of insulin with L-penetratin (RQIKIWFQNRRMKWKK) as a CPP demonstrated greater cognitive learning during the Morris water maze tests using a senescence-accelerated mouse-prone 8 (SAMP8) model than that with D-penetratin (rqikiwfqnrrmkwkk) [38]. In general, the endocytosis-inducing binding of cationic CPPs to anionic HSPGs is based on electrostatic interactions. In fact, L-CPPs and their corresponding D-CPPs showed similar binding affinities to HSPG [39]. Moreover, the membrane disturbance that induces direct translocation is based on hydrophobic and hydrophilic interactions between CPPs as surfactants and the membrane lipid bilayer or glycoproteins. It is not thought that CPP chirality matters with this translocation process, depending on the cases. However, L-CPPs were taken up greater than their corresponding D-CPPs in some types of cells, where D-CPPs persistently bound to HSPG but, in reverse, they were not so in some other types of cells [39]. D-CPPs do not interact largely with certain endogenous L-proteins, as they are protease-resistant. Accordingly, arbitrary L-CPPs might interact with endogenous L-proteins, different from degrative enzymes, at the time of cellular internalization. Therefore, intranasal L-penetratin with non-covalent insulin was suggested to interact with certain endogenous L-proteins that enhanced endocytosis and/or modulated tight junctions in the olfactory epithelium.

Nonetheless, insulin could be transferred through the paracellular pathway, depending on the conditions. Nasally administered insulin was delivered to the brain in vivo, although the mechanisms were unclear. It has to be qualified whether cycloheximide or other inhibitors inhibit or decrease insulin transportation. If transepithelial transportation is inhibited or decreased, it is implied that the paracellular pathway is involved. According to time-course analyses of insulin transfer, most transcytosis events through the transcellular pathway had completed within approximately 12 min after adding Alexa Fluor 568-labeled insulin to a confluent human adipose microvascular endothelial cell monolayer in an in vitro assay using total internal reflection fluorescence (TIRF) microscopy [31]. However, myosin light chain phosphorylation in rat intestinal tissue was significantly increased by 15 min and remained elevated at 45 min after intraluminal intestinal injection of 20 mM PIP peptide 640 in an in vivo assay. Furthermore, the level of blood glucose based on insulin through the paracellular pathway began to decrease at 15 min and reached a peak at 50 min after the intraluminal intestinal injection of 30 IU/kg insulin with 20 mM PIP peptide 640 in an in vivo assay using non-diabetic rats [36]. Thus, transepithelial events not only through the transcellular pathway but also through the paracellular pathway were so rapid that it was not possible to determine which pathway occurred from the time-course analyses. To reveal how insulin and other materials interact with the surface of the olfactory epithelium is essential for the design of delivered substances.

#### 2.6.2. Intranasal Insulin Trajectory to the Brain after Crossing the Olfactory Epithelium

Recently, it has been revealed that insulin plays a vital role in the prevention of cognitive impairment derived from AD [40]. It has been revealed that neurodegeneration in AD was caused not only by the accumulation of Aβ and tau but also by metabolic impairments in insulin signaling. Moreover, in the brain, insulin performs a crucial part in neurotrophic and neuromodulatory functions such as learning and memory as well as stem cell activation, cell growth, and synaptic plasticity [41]. Thus, insulin delivery into the brain can improve cognition in AD patients who tend to reduce CSF insulin levels by observation. Intranasal administration experiments were carried out using labeled insulin to ascertain insulin delivery into the brain in vivo. Alexa Fluor 647-labeled insulin after intranasal administration using mice moved from the olfactory mucosa to olfactory bulb along the extracellular olfactory nerve pathway for 30 min [42]. However, fluorescein isothiocyanate (FITC)-insulin after intranasal administration using rats moved from the nasal lamina propria to the brainstem along the extracellular perineurium for 30 min and epineural of the trigeminal nerve and then was located in the perivascular region of the cortex and widely in the brain, such as the cortex, cerebellum, and hippocampus [43]. Time-course analyses of insulin transfer took hours to reach from the olfactory epithelium to the olfactory bulb along the intracellular olfactory nerve pathway [44]. Thus, rapid transportation within minutes advocated the extracellular nerve pathway. However, intranasally labeled insulins demonstrated subtly different trajectories. Such different trajectories between Alexa Fluor 647-labeled insulin and FITC-insulin resulted from slight molecular modification to interact differently not only with endogenous proteins but also with themselves (Figure 6). Alexa Fluor 647-labeled insulin acted as a dimer, and FITC-insulin was used as the monomer [43]. Furthermore, the membrane permeability made a difference depending on the degree of FITC conjugate to insulin. The rank order of the permeability through the Madin-Darby canine kidney cell (MDCK) monolayer was mono-conjugate > unlabeled insulin > tri-conjugate [45]. Tri-conjugate is more hydrophobic than insulin. Accordingly, the internalization mechanism is not through passive lipoidal diffusion. Moieties introduced into insulin have influenced the features of all modified substances. Labeled-insulins modified to know the original insulin trajectory often demonstrated a trajectory different from that of unlabeled original insulin. This resembles the uncertainty principle of the German physicist Werner Heisenberg in quantum mechanics. As iodine is as large as the methyl group, it is thought that ^125^I-labeled insulin possesses the feature of unlabeled insulin. ^125^I-labeled insulin was detected in the olfactory bulb, hypothalamus, hippocampus, and cerebellum after intranasal administration in an in vivo assay using male CD-1 mice. ^125^I-labeled insulin distribution in the olfactory bulb was competitively inhibited by unlabeled insulin after intranasal administration [46]. Thus, insulin was delivered along the olfactory sensory nerve route. Moreover, in another experiment, intranasally administered ^125^I-labeled insulin was detected in the trigonal nerve, olfactory bulb, and olfactory epithelium 1 h later in an in vivo assay using mice [47,48].

#### 2.6.3. Intranasal Administration Using Nanoparticles

Cargo-loaded or cargo-containing nanoparticles have often been used as carriers in drug delivery systems and comprise micelles, liposomes, polymers, mesoporous silica, or metals. Biocompatible and biodegradable nanoparticles are desirable, particularly for the brain. Such a nanoparticle system could be applied for nose-to-brain delivery [49]. Almost 100% of insulin was released from insulin-encapsulating nanoparticle composed of chitosan/dextran sulfate (6:4 ratio, diameter of 320.55 nm) after approximately 12 h in in vitro test using a dialysis membrane [50]. However, at present, few insulin-loaded nanoparticles targeting the olfactory nerve route have been investigated. Chitosan is a biocompatible, biodegradable, and mucoadhesive cationic polymer that disrupts tight junctions (Figure 7). Nanoparticles composed of chitosan or chitosan/polyvinyl alcohol (PVA) loaded with insulin were developed. Nanoparticles composed of chitosan and insulin reduced plasma glucose levels more significantly than the corresponding nanoparticles composed of chitosan, PVA, and insulin in an in vivo assay after intranasal administration in female Wistar rats. Lowering the ratio of positively charged chitosan reduced mucoadhesiveness and tight junction opening consequently reducing insulin absorption [51]. The amount of insulin delivered into the brain is unknown. However, chitosan as an additive or carrier might enhance insulin delivery into the brain through the nose-to-brain route. Moreover, nanoparticles (diameter of 412.8 nm) composed of glyceryl monocaprylate-modified chitosan with a substitution degree of 12% (CS-GMC 12%), mixed with insulin at pH 4.5, enhanced the plasma glucose reduction compared to insulin alone after intranasal administration in an in vivo assay using conscious rats [52]. Chitosan–ZnO nanocomposition hydrogels can contain insulin in their porous structure. They decreased the blood glucose level by 50–65% of the initial level through intranasal administration using rats. This hydrogel formulation containing insulin transformed from the solution to viscous hydrogel in the nasal cavity at body temperature (37 °C). Such a hydrogel state can reduce the mucociliary clearance rate and release slowly insulin there [53]. Similarly, poly(dl-lactic acid-*co*-glycolic acid)/poly(ethylene glycol)/poly(dl-lactic acid-*co*-glycolic acid) (PLGA-PEG-PLGA) hydrogel (ICNPH) is the thermosensitive intelligent polymer, which is in a liquid state at room temperature (25 °C) and in a sol–gel transition state at body temperature (37 °C). The preparation of materials at room temperature is largely easy just by mixing liquid. Insulin-loaded chitosan nanoparticle/ICNPH (1:20 ratio) reduced retinal cell apoptosis greater compared to insulin alone in an in vivo assay using diabetic retinopathy model rats through a single subconjunctival injection [54]. The insulin delivery system using ICNPH might be also effective to intranasal administration, because appropriate gelatinization on the mucus layer of the epithelium delays mucociliary clearance. In addition, recently, albumin-based nanoparticles, as a colloidal carrier potent to load electrostatically or covalently with cargos, have been used for nose-to-brain substance delivery. Human serum albumin is a biodegradable, non-immunogenic protein (65–70 kDa). It is known that albumin transport pathways, mediated by gp60 or megalin/cubilin, function on the surface of endothelial and epithelial cells. Anti-inflammatory agents such as meloxicam (MEL) (Figure 8), a selective cyclooxygenase 2 (COX-2) inhibitor, are effective to neuroinflammation in AD. MEL-loaded albumin nanoparticles showed a higher cerebral concentration of MEL through intranasal administration in an in vivo assay using rats, compared to intravenous administration [55]. Thus, this system based on albumin-based nanoparticles can be applied for intranasal insulin delivery. Next, 3-aminophenylboronic-acid-conjugated dextran (Dex-PBA) nanoparticles (diameter of 329–348 nm) loaded with FITC-insulin were internalized into cells in an in vitro assay using Calu-3 cells as a nasal permeability model based on clathrin- and lipid raft/caveolae-dependent endocytosis. Dex-PBA nanoparticles loaded with insulin reduced the plasma glucose level more efficiently than insulin alone after intranasal administration in an in vivo assay using diabetic rats [56]. Furthermore, insulin-coated gold nanoparticles (INS-GNPs) (diameter of 20 nm) reportedly crossed the BBB by targeting InsR and were delivered into the brain after tail vein injection in an in vivo assay using Balb/C mice [57]. This INS-GNP formulation or its modified version might be applied for brain-targeted intranasal administration, not only based on receptor-mediated but also paracellular passage. Ideally, nanoparticles or dendrimers with as small diameters as possible might pass through the paracellular routes and cross the olfactory endothelium. However, endosomal diameters are 0.2–5 μm, 85–150 nm, 50–100 nm, and approximately 90 nm in the case of micropinocytosis, clathrin-dependent endocytosis, caveolae-dependent endocytosis, and clathrin- and caveolae-independent endocytosis, respectively [7]. When nanoparticles pass through the endocytosis-mediated transcellular routes, they must be contained in such endosomes. Otherwise, they must release insulin in the nasal mucosa of the olfactory or respiratory epithelium. In fact, insulin was delivered as cargo to the brain. However, in many cases, insulin was released from nanoparticles within the mucin layer. Successively, released insulin was transported into the epithelial cells across the membrane substantially as a vector by itself. Therefore, it is suggested that insulin can be used for the drug delivery of linked low molecular molecules.

#### 2.6.4. Clinical Trials of Intranasal Insulin Administration

It has been known for long that substances could be delivered from the olfactory epithelium to the brain. There are anatomical differences in the nasal cavity between species. The effectiveness of the direct nose-to-brain delivery in humans has not yet been validated, regardless of the success in animals. However, at present, a number of clinical trials using intranasal insulin have been performed. Searching clinical trial cases using the keywords “intranasal insulin” resulted in approximately 50 hits on the website of ClinicalTrials.gov [58]. Those trials that contained research design, methods, and results described in published articles were selected and are listed in Table 1. Although the clinical trials could be expected to demonstrate a positive correlation, it is worth verifying the relevant references on the field for further details [59,60,61,62,63,64,65,66,67].

### 2.7. Proposed Promising Methods of Intranasally Administered Drug Delivery into the Brain

#### 2.7.1. Possibility of Intranasal Insulin Conjugates with Low Molecular Agents as Cargo

In the olfactory epithelium, the paracellular pathway through loose tight junctions is more effective and simpler than the transcellular pathway through passive lipoidal diffusion or transcytosis. Thus, as designed delivery substances, middle-molecular-weight compounds are preferable to low- and large-molecular-weight compounds. Furthermore, suitable penetration into the nasal mucosa and the avoidance of mucociliary clearance are also required. Nonetheless, intranasally administered brain-targeted drug delivery should be foreseen to pass through not only the olfactory nerve but also the trigeminal nerve due to mucociliary clearance and off-target adhesion to the respiratory region instead of the olfactory region. In fact, in vivo observations suggested that intranasal insulin could also pass through the trigeminal nerve. It might be easier not for hydrophobic small molecular compounds but for insulin to pass through the trigeminal nerve perineuronal space than to cross the continuous and fenestrated endothelium of blood vessels from the basolateral to the apical cell membrane based on transcytosis at the lamina propria of the respiratory epithelium, if the trigeminal nerve perineuronal space is open to extracellular fluid, similar to how the conduit interstitially formed by the olfactory ensheathing cells surrounding the olfactory sensory neuron is open to CSF. This hypothesis might be supported by the fact that less than 3% of intranasally administered ^125^I-insulin was detected in the serum of CD-1 mice [46]. Therefore, insulin, with or without additives such as tight junction openers or CPPs, as time-proven substances, should be among the design candidates of intranasally administered substances that are subject to be delivered into the brain as vectors and carriers. In fact, Alexa Fluor 647-labeled insulin and FITC-insulin were intranasally delivered into the brain. Thus, this strategy using insulin as a carrier and vector could deliver small molecular compounds from the nose to the brain, although it has not been conducted yet. On the other hand, as the similar conjugate concept, it was reported that a hybrid insulin peptide composed of an insulin C-peptide fragment fused to a peptide from chromogranin A that was a T cell antigen in human type 1 diabetes, loaded on poly(lactide-*co*-glycolide) (PLG) nanoparticle, was developed and evaluated to exhibit the re-educative activity to T cells in an ex vivo study using the spleen and pancreas from euthanized mice [68].

#### 2.7.2. Design of Intranasal Insulin Conjugates with Low Molecular Agents

Insulin has several functional groups that could be tethered with cargos through a suitable linker. In the case of human insulin (Figure 4), the amino groups are ^1^Gly in the A-chain and ^1^Phe, ^22^Arg, and ^29^Lys in the B-chain. The reactivity of ^1^Phe in the B-chain is the highest among the amino groups of insulin [45]. The hydroxyl groups are ^9^Ser, ^14^Tyr, and ^19^Tyr in the A-chain and ^9^Ser and ^27^Thr in the B-chain. The carboxyl groups are ^5^Glu, ^17^Glu, and ^21^Asn in the A-chain and ^13^Glu, ^21^Glu, and ^30^Thr in the B-chain. In fact, modified insulin conjugates with Alexa Fluor 647- or FITC-conjugated amino groups were shown to be delivered into the brain via the intranasal route. Ideally, linkers should be easily and timely cleaved after transportation across the olfactory epithelium to release the cargo there. When the conjugates pass through the paracellular pathway after intranasal administration, such a transportation process rapidly proceeds and is free from enzymatic metabolism, similar to that in the liver or serum. Accordingly, the ester bond might be sufficient for the linker function. Galantamine (Figure 8), an acetylcholinesterase inhibitor, was approved by the Food and Drug Administration for the treatment of AD. It might be absorbed into the brain by the H^+^/OC antiporter on the apical membrane of capillary endothelial cells at the BBB due to *N*-containing structural similarity to morphine, which is an H^+^/OC antiporter substrate [69,70]. However, collision between galantamine and the transporters is restricted in the systemic circulation. Galantamine-loaded solid lipid nanoparticles (diameter of < 100 nm) were developed by the microemulsification method and showed enhanced memory restoration capability compared to galantamine alone in an in vivo assay using orally administered cognitive deficit rat models [71]. Galantamine has a hydroxyl group. Therefore, a galantamine–insulin conjugate with an esterified linker could be a potent substance to elicit activity in the brain after intranasal administration. Moreover, insulin itself improved learning and memory. Thus, a galantamine–insulin conjugate could synergistically elicit ameliorating effects. Furthermore, Hes1 dimer inhibitors (Figure 8) could induce neural stem cell (NSC) differentiation [72]. Conjugates introduced to insulin with suitable linkers have the potential to be used for the treatment of neurodegenerative diseases through nasal administration. Histone deacetylase 3 (HDAC3) inhibitors and SIRT2 inhibitors (Figure 8) could also be therapeutic agents for neurodegenerative diseases [73,74]. Intranasally administered conjugates of insulin with suitable linkers might be delivered into the brain, respectively. Moreover, γ-secretase inhibitors (Figure 8) reduced Aβ production [75]. Thus, their conjugates with insulin could provide remedy for AD through nasal administration.

The insulin-degrading enzyme (IDE) cleaves both insulin chains and Aβ (1–40) at several positions (Figure 4) [76]. Insulin, linked with cargo at the amino group of ^1^Phe in the B-chain IDE-binding region, might not competitively interrupt the IDE-mediated Aβ degradation, and it might bind to InsRs to activate them, although the first leucine-rich repeat (L1) domain and the InsR αCT interacts with the insulin B-helix [77]. ^1^Gly, ^5^Gln, ^19^Tyr, and ^21^Asn in the A-chain and ^10^His, ^16^Tyr, ^23^Gly, ^24^Phe, and ^25^Phe in the B-chain are involved in InsR binding [78]. Insulin conjugates linked with cargo at the amino group of ^22^Arg near the α-helix (^9^Ser-^19^Cys) on the B-chain might not bind to InsR. Two α-helix structures on the A-chain are formed at ^2^Ile-^8^Thr and ^13^Leu-^19^Tyr. As insulin could be degraded not only by IDE but also by other enzymes, such as protein disulfide isomerase [79], insulin derivatives will not remain in the brain for a while due to metabolic clearance. Insulin derivatives and their metabolites are removed by the glymphatic system. The insulin conjugate strategy is considered to be a low-toxicity drug delivery approach. Nevertheless, the optimization of the activity and the delivery of the designed insulin conjugates must be fine-tuned based on repeated experimental results. After the establishment of the proof-of-concept, insulin as a carrier might be subject to replacement with other materials such as biocompatible and biodegradable polymers for better stability and product cost. From the findings about mucus layer permeation of insulin-loaded nanoparticles at the epithelium, insulin conjugates with bioactive substances particularly loaded on mucoadhesive nanoparticles and hydrogels would be distributed to the brain through a nose-to-brain route greater than insulin conjugates with bioactive substances alone.

## 3. Conclusions

It has been known for long that a direct connection exists between the nose and the brain through the cribriform plate. Thus, direct nose-to-brain drug delivery is a promising potential strategy to ameliorate CNS diseases such as AD and PD. However, despite the expectation, intranasal administration to the brain depends not only on the olfactory nerve route but also on the respiratory route, which is probably due to the narrowness of the human olfactory region in the nose cavity, location at the top of the nasal cavity, and mucociliary clearance. This fact has to be accepted in the discovery and development of drugs delivered through the nose-to-brain route. Intranasally administered insulin was delivered to the brain through both routes (Table 2). Nonetheless, it has been revealed that nasal insulin improved AD dementia in animal models. Thus, a large number of nasal insulin delivery-based clinical trials have been carried out in order to investigate cognitive function improvement. Inherently, in CNS disease-related drug development, the BBB makes substance delivery into the brain difficult. Thus, in order to bypass the BBB, the intranasal route to the brain could be an alternative approach. In fact, intranasally administered insulin and derivatives such as Alexa Fluor 647-labeled insulin and FITC-insulin were successfully delivered into the brain. Taking advantage of insulin as a vector and/or carrier is undeniably beneficial for delivering cargo, using either cleavable covalent or non-covalent linkers, into the brain through the nose-to-brain route. Olfactory epithermal cells could be injured by outer materials and regenerate with at a high turnover rate. As a result, tight junctions at such damaged sites are apt to be loose and leaky enough to allow macromolecules, such as insulin conjugates with low-molecular-weight compounds, pass through the paracellular pathway and be delivered into the brain. This strategy would be a useful method for drug delivery into the brain. The aromaticity of planar ring compounds is strictly determined by 4n + 2 electrons based on Hückel’s rule. However, the directionality of intranasally administered compounds is unpredictable and likely to be determined by the molecular disorder based on Boltzmann’s principle of statistical mechanics. One molecule might still move as intended, although the case of more molecules would be different. Thus, research effort should focus on increasing the precision of intranasal drug delivery targeting to the brain by designing functionalized substances and developing devices such as aerosols or a special type of epithelium patch. Consequently, the establishment of direct drug delivery through the nose-to-brain route with or without nanoparticles using nano drug delivery systems could potentially contribute to healing CNS diseases.

## Figures and Tables

**Figure 1 molecules-25-05188-f001:**
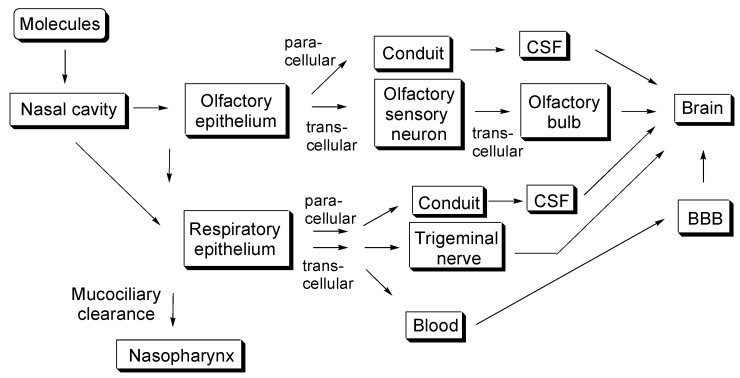
Schematic overview of several molecular pathways after intranasal drug administration. CSF stands for cerebrospinal fluid. BBB stands for blood–brain barrier.

**Figure 2 molecules-25-05188-f002:**
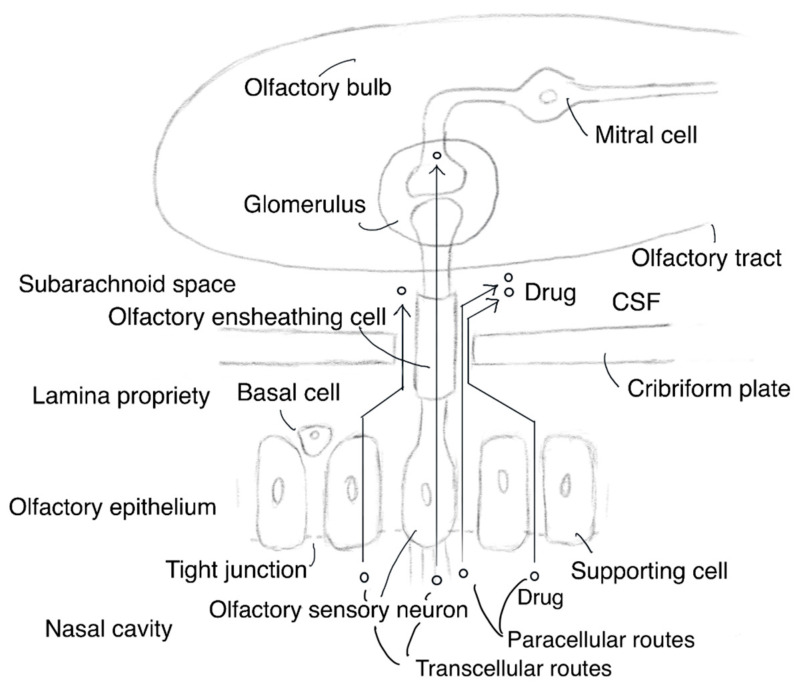
Structure of the olfactory epithelium and molecular distribution pathways. CSF stands for cerebrospinal fluid.

**Figure 3 molecules-25-05188-f003:**
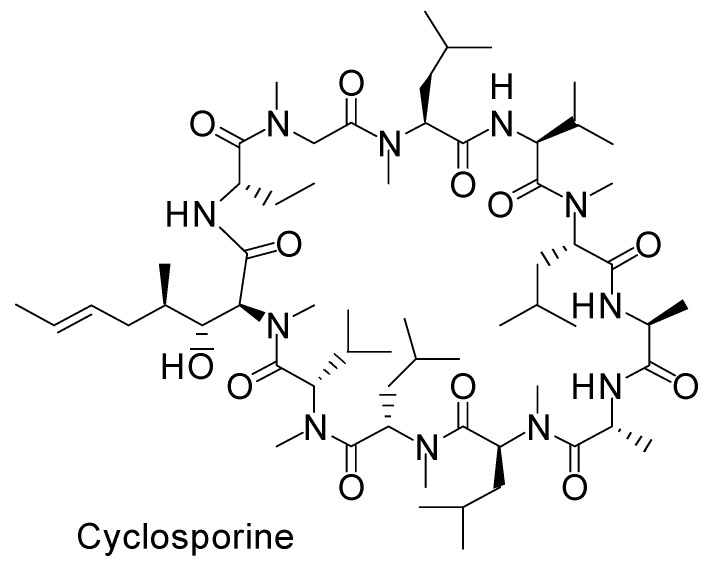
*N*-Methylated cyclic structure of cyclosporine.

**Figure 4 molecules-25-05188-f004:**
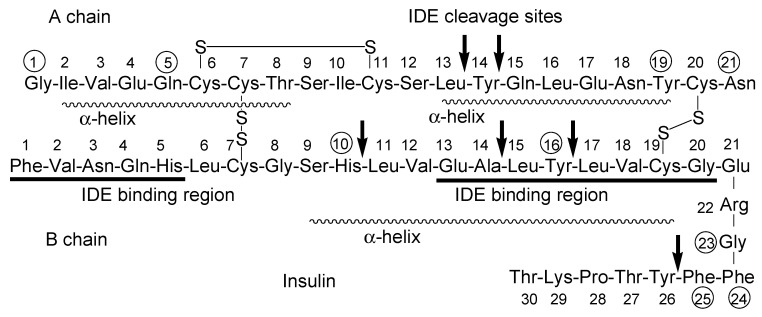
Structure of human insulin. Arrows indicate IDE cleavage sites. IDE binding regions are underlined. Circled numbers indicate amino acids involved in insulin receptor (InsR) binding. α-Helix regions are indicated with broken lines. IDE stands for insulin-degrading enzyme.

**Figure 5 molecules-25-05188-f005:**
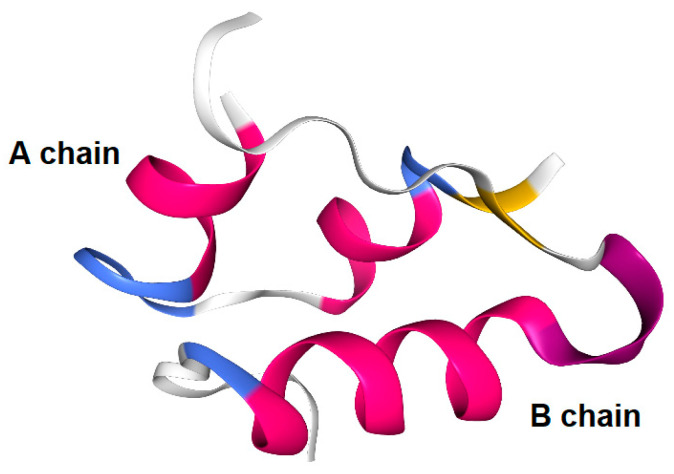
Human insulin X-ray crystal structure (3I40).

**Figure 6 molecules-25-05188-f006:**
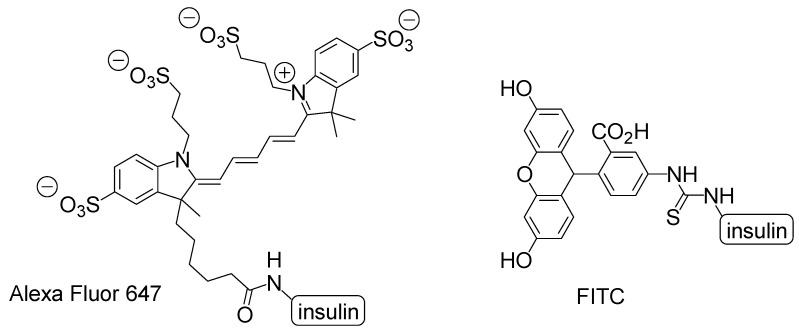
Structures of imaging dyes Alexa Fluor 647 and fluorescein isothiocyanate (FITC) introduced to the amino group of insulin.

**Figure 7 molecules-25-05188-f007:**
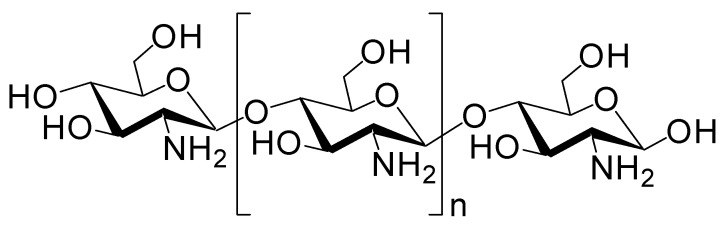
Structure of chitosan.

**Figure 8 molecules-25-05188-f008:**
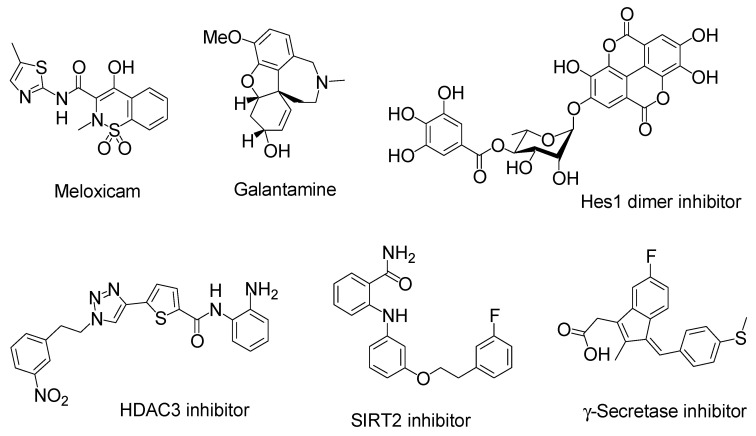
Structures of neurodegenerative disease modulators. HDAC stands for histone deacetylase. SIRT stands for sirtuin.

**Table 1 molecules-25-05188-t001:** Representative clinical trials focusing on intranasal insulin delivery.

#	Administration	Condition or Disease	Sponsor	Phase	Study Start Date	Study Completion Date	ClinicalTrials.gov Identifier	Reference
(i)	Intranasal Insulin	Mild Cognitive Impairment Alzheimer’s Disease	University of Washington	Phase 2	June 2006	December 2011	NCT00438568	[59]
(ii)	Intranasal Insulin	Type 2 Diabetes Mellitus	Beth Israel Deaconess Medical Center	Phase 2	May 2010	April 2013	NCT01206322	[60]
(iii)	Intranasal Insulin	Alzheimer’s Disease Mild Cognitive Impairment	University of Washington	Phase 2	March 2011	December 2012	NCT01547169	[61]
(iv)	Intranasal Insulin	Alzheimer’s Disease Mild Cognitive Impairment	Wake Forest University Health Sciences	Phase 2	November 2011	March 2015	NCT01595646	[62]
(v)	Intranasal Insulin	Parkinson’s Disease Multiple System Atrophy	Peter Novak	Phase 2	February 2014	September 2015	NCT02064166	[63]
(vi)	Intranasal Insulin	Hyperlipidemia	University Health Network, Toronto	Phase 2 Phase 3	April 2016	April 2017	NCT03141827	[64]
(vii)	Intranasal Insulin	Type 2 Diabetes Mellitus	Beth Israel Deaconess Medical Center	Phase 2 Phase 3	July 2015	June 2020	NCT02415556	[65]
(viii)	Intranasal Insulin	Bipolar Disorder	University Health Network, Toronto	Phase 3	May 2006	March 2009	NCT00314314	[66]
(ix)	Intranasal Insulin	Diabetes	German Diabetes Center	Phase 4	August 2011	June 2018	NCT01479075	[67]

**Table 2 molecules-25-05188-t002:** Summary of the delivery approaches of insulin and its derivatives described in this review.

#	Compound	Assay	Additives/ Components	Administration	Cells/ Animals	Results	References
(i)	Insulin	In vitro	-	-	T-84 cell monolayer	Tight junction opening	[32]
(ii)	Insulin	In vitro	Octylglucoside	-	T-84 cell monolayer	Paracellular permeation enhancement	[34]
(iii)	Insulin	In vivo	PIP peptide 640	Intraluminal intestinal injection	Rats	Blood glucose level reduction	[36]
(iv)	Cy3-labeled insulin	In vivo	PIP peptide 640	Intraluminal intestinal injection	Rats	Detection in the paracellular route	[36]
(v)	Insulin	In vivo	l-Penetratin	Intranasal injection	Mice	Cognitive learning enhancement	[38]
(vi)	Alexa Fluor 568-labeled insulin	In vitro	-	-	Adipose microvascular endothelial cells	Transcytosis	[31]
(vii)	Alexa Fluor 647-labeled insulin	In vivo	-	Intranasal injection	Mice	Delivery to the olfactory bulb	[42]
(viii)	FITC-insulin	In vivo	-	Intranasal injection	Rats	Delivery to brain	[43]
(ix)	^125^I-labeled insulin	In vivo	-	Intranasal injection	Mice	Delivery to trigonal nerve and olfactory bulb	[46,47,48]
(x)	Insulin loaded on nanoparticle	In vitro	Chitosan/dextran sulfate	-	Cellulose acetate membranes	Insulin release from nanoparticle	[50]
(xi)	Insulin loaded on nanoparticle	In vivo	Chitosan/PVA	Intranasal injection	Rats	Blood glucose level reduction	[51]
(xii)	Insulin loaded on nanoparticle	In vivo	Chitosan /GMC	Intranasal injection	Rats	Blood glucose level reduction	[52]
(xiii)	Insulin loaded on nanoparticle	In vivo	Chitosan–ZnO/ hydrogel	Intranasal injection	Rats	Blood glucose level reduction	[53]
(xiv)	Insulin loaded on nanoparticle	In vivo	PLGA–PEG–PLGA/ hydrogel	Subconjunctival injection	Rats	Retinal cell apoptosis reduction	[54]
(xv)	Insulin loaded on nanoparticle	In vivo	Dex–PBA	Intranasal injection	Rats	Blood glucose level reduction	[56]
(xvi)	Insulin loaded on gold nanoparticle	In vivo	Gold	Tail vein injection	Mice	BBB penetration	[57]
(xvii)	Insulin	In vivo	-	Intranasal injection	Humans	Positive correlations with clinical trials	Table 1 [59,60,61,62,63,64,65,66,67]
(xviii)	Hybrid insulin peptide loaded on nanoparticle	Ex vivo	Chromogranin A PLG nanoparticle	-	Spleen and pancreas from mice	Re-educative activity to T cells	[68]
(xix)	Insulin conjugate	In vivo	Galantamine	Intranasal injection	-	Under analysis in Tashima lab	-
(xx)	Insulin conjugate	In vivo	Hes1 dimer inhibitor	Intranasal injection	-	Under analysis in Tashima lab	[72]
(xxi)	Insulin conjugate	In vivo	HDAC3 inhibitor	Intranasal injection	-	Under analysis in Tashima lab	[73]
(xxii)	Insulin conjugate	In vivo	SIRT2 inhibitor	Intranasal injection	-	Under analysis in Tashima lab	[74]
(xxiii)	Insulin conjugate	In vivo	γ-Secretase inhibitor	Intranasal injection	-	Under analysis in Tashima lab	[75]

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
