# Peer review of "Shortcut Approaches to Substance Delivery into the Brain Based on Intranasal Administration Using Nanodelivery Strategies for Insulin"

_molecules, 2020, doi:10.3390/molecules25215188_

Round 1

Reviewer 1 Report

The aims of the review proposed by Toshihiko Tashima are to describe the interest of the intranasal administration route to bypass the BBB and to permit efficient delivery of insulin into CNS, in particular to cure CNS-related diseases such as Alzheimer's disease. Some promising strategies in terms of insulin-based cargos design are described, and are related to their associated transport pathway.

Such a review is interesting for the scientific community, and well documented. Some corrections should be performed to improve its final form.

  1. A carreful care should be done on the format of the text to exactly correspond to the Molecules template, in particular the space between lines.
  2. The title should be modified to best define the objectives of the review.
  3. Please, avoid sentences with "I" (l.21, l.59, l.65, l.367, ...) and personal points of view as "despite my expectation" (l.565)
  4. In the abstract, l.8, why is a plurial form used for insulin ?
  5. A figure should be added to easier differentiate the transport pathways accross the BBB, and including the lymphatic system contribution, which is interestingly described in 2.5 part. What does "lipoidal" mean in the "passive lipoidal diffusion" (used l.271)
  6. Please, include the abbreviations significance into the titles of the figures, e.g. CSF (Figure 1), IDE (Figure 4)
  7. The resolution of the Figure 2 should be improved. The figure 5 should be suppressed: what about its utility ? The title of the Figure 6 must be completed. The term "chitosan" must be suppressed on  the Figure 7 and its chemical formula should be more general
  8. l.157: the sentence appears unclear. Please give examples of intranasal drugs already introduced in the market and able to reach the CNS
  9. l.287: to which systems do the "certain types of insulin transportation systems" refer ? It is unclear
  10. l.310: what is the particularity of the PIP ? Please, best introduce these peptides
  11. What is the formulation type used to administrate insulin in examples that are given in the 2.6.2 part ? It would be useful to precise that since it can impact the transport mode and its efficiency as explained with nanoparticles (2.6.3)
  12. Subtitles should be added in 2.7 part to highlight the different strategies. Moreover, it is not clear if these strategies (in particular with galantamine-insulin complex) have been already developed and assayed, or if it is just a proposal of the author. It should be clarified within the text. In consequence, in Figure 8, not only the modulators should be despicted but also the insulin-based conjugates

Author Response

I appreciate reviewers’ comments, suggestions, and rectifications very much. According to these, I revised the manuscript as below.

Response to Reviewer 1 Comments

The aims of the review proposed by Toshihiko Tashima are to describe the interest of the intranasal administration route to bypass the BBB and to permit efficient delivery of insulin into CNS, in particular to cure CNS-related diseases such as Alzheimer's disease. Some promising strategies in terms of insulin-based cargos design are described, and are related to their associated transport pathway.

Such a review is interesting for the scientific community, and well documented. Some corrections should be performed to improve its final form.

1.A carreful care should be done on the format of the text to exactly correspond to the Molecules template, in particular the space between lines.

Response 1: I correct the format in particular the space between lines.

2.The title should be modified to best define the objectives of the review.

Response 2: I added “nanodelivery strategies for insulin” in title.

3.Please, avoid sentences with "I" (l.21, l.59, l.65, l.367, ...) and personal points of view as "despite my expectation" (l.565)

Response 3: I modified the sentences. I replaced “I wanted to know” with “It has to be qualified”. I replaced “despite my expectation” with “despite the expectation”. However, three I’s remain in abstract and introduction, because I must run a small affiliation group. Thus, I must make a presence. Basically, I am alone. This manuscript was written only by me. Former articles like this style used first-person pronouns. Please be understanding of this.

4.In the abstract, l.8, why is a plurial form used for insulin ?

Response 4: Labeled insulins are derivative forms such as Cy3-labeled insulin, Fluor 647-labeled insulin, and FITC-insulin. This expression was checked by English proof service (editage).

5.A figure should be added to easier differentiate the transport pathways accross the BBB, and including the lymphatic system contribution, which is interestingly described in 2.5 part. What does "lipoidal" mean in the "passive lipoidal diffusion" (used l.271)

Response 5: Passive lipoidal diffusion is used to distinguish from paracellular passive diffusion. The term "lipoidal" means lipid bilayer that is transported by hydrophobic compounds.

6.Please, include the abbreviations significance into the titles of the figures, e.g. CSF (Figure 1), IDE (Figure 4)

Response 6: I added “CSF stands for cerebrospinal fluid. BBB stands for blood-brain barrier.” In Figure 1, “CSF stands for cerebrospinal fluid.” in Figure 2, “IDE stands for Insulin degrading enzyme.” In Figure 4, and “HDAC stands for histone deacetylase. SIRT stands for sirtuin.” in Figure 8.

7.The resolution of the Figure 2 should be improved. The figure 5 should be suppressed: what about its utility ? The title of the Figure 6 must be completed. The term "chitosan" must be suppressed on the Figure 7 and its chemical formula should be more general

Response 7: I added “transcellular routes” and “paracellular routes” in Figure 2. I made figure 5 smaller. Insulin 3-dimentional structure provide the information to design delivered compounds. I added “introduced to the amino group of insulin” in Figure 6. I made figure 7 smaller. The term "chitosan" was deleted.

8.l.157: the sentence appears unclear. Please give examples of intranasal drugs already introduced in the market and able to reach the CNS

Response 8: I added “, such as oxytocin (1007 Da) and desmopressin (1069 Da)”.

9.l.287: to which systems do the "certain types of insulin transportation systems" refer ? It is unclear

Response 9: “Plausible routes are divided into transcellular pathways, including RMT, other types of endocytosis/exocytosis such as macropinocytosis, passive diffusion, direct translocation based on CPPs, and paracellular pathway.”

10.l.310: what is the particularity of the PIP ? Please, best introduce these peptides

Response 10: The cited patent described the particularity of the PIP with respect to induction of “myosin light chain phosphorylation to enhance paracellular transport by opening tight junctions”.

I added “PIP peptide 640 was initially found as an inhibitor of the interaction between protein phosphatase 1 (PP1) and its regulator CPI-17.”.

However, I noticed that the expression between the patent and the literature was different. Thus, I corrected PIP peptide 640 and 250 according to the literature. I replaced “PIP peptide 250 (RKAKYQYRRK) and PIP peptide 640 (RRVEVKYDRR) )” with “PIP peptide 250 (rrfkvktkkrk -NH2) as D-form and PIP peptide 640 (rrdykvevrr-NH2) as D-form”.

11.What is the formulation type used to administrate insulin in examples that are given in the 2.6.2 part ? It would be useful to precise that since it can impact the transport mode and its efficiency as explained with nanoparticles (2.6.3)

Response 11: (in the 2.6.2 part) To investigate the nasal insulin trajectory, labelled insulins were used. That is, Alexa Fluor 647-labeled insulin, FITC-insulin, and 125I-labeled insulin. These were not with nanoparticle, but themselves alone.

(in the 2.6.2 part) In many cases insulin was released from nanoparticle within mucin layer. Successively, released insulin was transported into the epithelial cells across the membrane. Thus, it is useful to know nasal insulin trajectory in nose-to-brain. Moreover, the nasal insulin trajectory gave us the hints to design molecules. Nanoparticles assisted insulin delivery by mucoadherance and protection from degradation.

12.Subtitles should be added in 2.7 part to highlight the different strategies. Moreover, it is not clear if these strategies (in particular with galantamine-insulin complex) have been already developed and assayed, or if it is just a proposal of the author. It should be clarified within the text. In consequence, in Figure 8, not only the modulators should be despicted but also the insulin-based conjugates

Response 12: I added subtitles such as “Possibility of intranasal insulin conjugates with low molecular agents as cargo”

and “Design of intranasal insulin conjugates with low molecular agents”.

The structures of the insulin-based conjugates are not shown because of patents by not only me but also some inspired readers.

I added “although it has not been conducted yet. On the other hand, as the similar conjugate concept, it was reported that a hybrid insulin peptide composed of an insulin C-peptide fragment fused to a peptide from chromogranin A that was a T cell antigen in human type 1 diabetes, loaded on poly(lactide-co-glycolide) (PLG) nanoparticle, was developed and evaluated to exhibit the re-educative activity to T cells in an ex vivo study using spleen and pancreas from euthanized mice.”

That is all.

Reviewer 2 Report

Line 60 - adviced to mentione the significance of nanoparticles/nano drug delivery systems for this purpose

Section 2.3 deals with the development of substance delivery to the brain through intranasal administration, but it's not clear, what type of carrier modifications can support nose-to-brain delivery.

Line 174 describes the negative effect of th mucociliar clearance; it would be important to mention some techniques or excipients available to increase the residence time of the drug in the nasal mucosa.

Line 385 the hypothetic mechanism of action of insulin against the cognitive impairment is missing

Section 2.6.3 albumin based nanoparticles have also significant impact in nose-to-brain delivery, I advise to add them to the list

Author Response

I appreciate reviewers’ comments, suggestions, and rectifications very much. According to these, I revised the manuscript as below.

Response to Reviewer 2 Comments

Line 60 - adviced to mentione the significance of nanoparticles/nano drug delivery systems for this purpose

Response 1: I added “Furthermore, nanoparticles based on nanodelivery strategies have been extraordinarily used for membrane permeation.” and “with or without nanoparticles using nano drug delivery systems”.

Section 2.3 deals with the development of substance delivery to the brain through intranasal administration, but it's not clear, what type of carrier modifications can support nose-to-brain delivery.

Response 2: I added “At present, drug delivery using nanoparticle systems has been developed. When nanocarriers are used for nose-to-brain drug delivery to enhance availability, they should properly exhibit protection from degradation, mucoadherance, transparent into mucus layer, cargo release at appropriate sites, and rapid transepithelium of cargo released from or contained in carrier.”.

Line 174 describes the negative effect of th mucociliar clearance; it would be important to mention some techniques or excipients available to increase the residence time of the drug in the nasal mucosa.

Response 3: I added “As one of methods to increase the residence time of the drug in the nasal mucosa, it is suggested that mucoadhesive nanocarriers composed of cationic materials electrostatically attach to anionic mucosa and subsequently turn into a gel induced by body temperature there to release gradually the cargos. Chitosan and hydrogels might play such role.”.

Line 385 the hypothetic mechanism of action of insulin against the cognitive impairment is missing

Response 4: I added “It has been revealed that neurodegeneration in AD was caused not only by the accumulation of Aβ and tau but also by metabolic impairments in insulin signaling. Moreover, in the brain, insulin performs a crucial part in neurotrophic and neuromodulatory functions such as learning and memory as well as stem cell activation, cell growth, and synaptic plasticity. Thus, insulin delivery into the brain can improve cognition in AD patients who tend to reduce CSF insulin levels by observation.”

Section 2.6.3 albumin based nanoparticles have also significant impact in nose-to-brain delivery, I advise to add them to the list

Response 5: I add “In addition, recently, albumin-based nanoparticles, as colloidal carrier potent to load electrostatically or covalently with cargos, have been used for nose-to-brain substance delivery. Human serum albumin is a biodegradable, non-immunogenic protein (65–70 kDa). It is known that albumin transport pathways, mediated by gp60 or megalin/cubilin, function on the surface of endothelial and epithelial cells. Anti-inflammatory agents such as meloxicam (MEL), a selective COX-2 inhibitor, are effective to neuroinflammation in AD. MEL-loaded albumin nanoparticles showed higher cerebral concentration of MEL through intranasal administration in an in vivo assay using rats, compared to intravenous administration. Thus, this system based on albumin-based nanoparticles can be applied for intranasal insulin delivery.”

That is all.

Reviewer 3 Report

This article intended to review the role of insulin in intranasal route approaches for brain delivery.  The theme is interesting and timely, since drug delivery to the brain is very challenging and hinders the success of many therapeutic drugs. Nose-to-brain delivery is a promising strategy and is currently extensively studied due to the ability to bypass the blood-brain barrier and to increase drug bioavailability in the brain.

In my opinion, the manuscript should not be accepted for publication as it is. The article is not well structured, presenting several concepts in an unconnected manner. Also, this article is more appropriate for a Perspective or an Opinion article rather than a review, since it presents a concept from a personal viewpoint proposing a new approach in a speculative way. A review should provide the most recent progress made in a research field. However, the author did not review the research progress on the use of insulin as strategy for intranasal administration as is mentioned in the abstract and introduction. In fact, the author only proposes the use of insulin for this purpose.

Below the author could find a list of suggestions to improve the quality of the manuscript.

I suggest avoiding using first-person pronouns in the manuscript.

Lines 84 to 98: this text is outside the scope of this review. Please remove. I also suggest changing the title of section 2.1 to “Anatomical features of the nasal cavity”, accordingly.

Lines 152 to 170; 182-195: Although the information given here is well known, literature references should be added to support it.

Figure 2: please add a legend indicating the paracellular and the transcellular routes within the figure.

Lines 269-271: information about the immunosuppressor cyclosporine seems out of context here. Although the author used it as an example of a drug that is passively transported unlike insulin, please rewrite de rephrase for clarification.

Figure 6: this figure is not relevant for the article. Please remove.

Lines 421-454: The author cited and discussed 4 research works on nanoparticles for insulin intranasal delivery. Several other works have been published in this field recently. The author should explain its exclusion/inclusion criteria for the cited works. Also, the presentation of descriptive lists of previously published work should be avoided. A critical vision of the state of the art is the main demand from the scientific community. It would be interesting if the author could add a discussion to give a critical opinion on the developed works. For example, the author could discuss the need/advantages of using nanoparticles for insulin delivery, as several evidences report the ability of free insulin to reach the brain through intranasal route.

However, is not clear what the purpose of this section as, in these cited works, insulin was delivered as cargo in the nanoparticles, and not used as the vector/targeting moiety for enhanced intranasal delivery that was the main purpose of this review.

Lines 510-528: The author proposes here the use of insulin conjugates to improve the brain delivery of some molecules with proven therapeutic effects. The author did not review the recent research progress of these strategies, but only presented this as a potential strategy. Thus, the author should reformulate the title of the article, as well as the abstract and the last lines of the introduction, which states that the purpose of this review is to discuss the use of insulin as a vector for intranasal delivery. I also suggest the author to reorganize the manuscript to highlight the nanodelivery strategies for insulin (section 2.6.3) adding other research works in this filed, and to further explore table 2. This table is very interesting and summarizes different studied delivery approaches for insulin (intranasal and other delivery routes). However, it is not well explored and discussed in the manuscript.

References list: Reference 71: the article title is wrong. Correct title is: “Monomeric Insulins and Their Experimental and Clinical Implications”

Author Response

I appreciate reviewers’ comments, suggestions, and rectifications very much. According to these, I revised the manuscript as below.

Response to Reviewer 3 Comments

This article intended to review the role of insulin in intranasal route approaches for brain delivery. The theme is interesting and timely, since drug delivery to the brain is very challenging and hinders the success of many therapeutic drugs. Nose-to-brain delivery is a promising strategy and is currently extensively studied due to the ability to bypass the blood-brain barrier and to increase drug bioavailability in the brain.

In my opinion, the manuscript should not be accepted for publication as it is. The article is not well structured, presenting several concepts in an unconnected manner. Also, this article is more appropriate for a Perspective or an Opinion article rather than a review, since it presents a concept from a personal viewpoint proposing a new approach in a speculative way. A review should provide the most recent progress made in a research field. However, the author did not review the research progress on the use of insulin as strategy for intranasal administration as is mentioned in the abstract and introduction. In fact, the author only proposes the use of insulin for this purpose.

Response 1: I replaced “review” with “perspective review”.

Below the author could find a list of suggestions to improve the quality of the manuscript.

I suggest avoiding using first-person pronouns in the manuscript.

Response 2: I modified the sentences. I replaced “I wanted to know” with “It has to be qualified”. I replaced “despite my expectation” with “despite the expectation”. However, three I’s remain in abstract and introduction, because I must run a small affiliation group. Thus, I must make a presence. Basically, I am alone. This manuscript was written only by me. Former articles like this style used first-person pronouns. Please be understanding of this.

Lines 84 to 98: this text is outside the scope of this review. Please remove. I also suggest changing the title of section 2.1 to “Anatomical features of the nasal cavity”, accordingly.

Response 3: This manuscript was written for medicinal chemists, including me, that should design and produce drugs. They know much about organic synthesis, however, are unfamiliar with biochemistry. So, I submitted it to “Molecules”. Through these phrases, the structure and system of nose-to-brain was shown. I am sorry that I want to leave these phrases. Please be understanding of this. Moreover, I replaced “Anatomical features of tissues involved in the sense of smell” with “Anatomical features of the nasal cavity”

Lines 152 to 170; 182-195: Although the information given here is well known, literature references should be added to support it.

Response 4: I added “(5,6)” for Lines 152 to 170 and “(4)” for 182-195.

Figure 2: please add a legend indicating the paracellular and the transcellular routes within the figure.

Response 5: I added “paracellular route” and “transcellular route” in Figure 2.

Lines 269-271: information about the immunosuppressor cyclosporine seems out of context here. Although the author used it as an example of a drug that is passively transported unlike insulin, please rewrite de rephrase for clarification.

Response 6: I added “although cyclosporine is a type of peptide” and “due to insulin size and non-N-methylated hydrophilicity”.

Figure 6: this figure is not relevant for the article. Please remove.

Response 7: The moieties introduced to insulin have impact on pharmacokinetics. Actually, Alexa Fluor 647 and FITC showed different trajectories across the epithelium after intranasal administration. Thus, I think that the structures of Alexa Fluor 647 and FITC in Figure 6 should be shown.

Then I added “Furthermore, the membrane permeability made a difference depending on degree of FITC conjugate to insulin. The rank order of the permeability through the MDCK monolayer was mono-conjugate > unlabeled insulin > tri-conjugate. Tri-conjugate is more hydrophobic than insulin. Accordingly, the internalization mechanism is not through passive lipoidal diffusion.” and “The reactivity of 1Phe in the B-chain is the highest among the amino groups of insulin”.

Lines 421-454: The author cited and discussed 4 research works on nanoparticles for insulin intranasal delivery. Several other works have been published in this field recently. The author should explain its exclusion/inclusion criteria for the cited works. Also, the presentation of descriptive lists of previously published work should be avoided. A critical vision of the state of the art is the main demand from the scientific community. It would be interesting if the author could add a discussion to give a critical opinion on the developed works. For example, the author could discuss the need/advantages of using nanoparticles for insulin delivery, as several evidences report the ability of free insulin to reach the brain through intranasal route.

However, is not clear what the purpose of this section as, in these cited works, insulin was delivered as cargo in the nanoparticles, and not used as the vector/targeting moiety for enhanced intranasal delivery that was the main purpose of this review.

Response 8: I added “In fact, insulin was delivered as cargo to the brain. However, in many cases insulin was released from nanoparticle within mucin layer. Successively, released insulin was transported into the epithelial cells across the membrane substantially as vector by itself. Therefore, it is suggested that insulin can be used for drug delivery of linked low molecular molecules.”

Furthermore, I added the other research work in section 2.6.3.

I added “Almost 100% of insulin was released from insulin-encapsulating nanoparticle composed of chitosan/dextran sulfate (6 : 4 ratio, diameter of 320.55 nm) after approximately 12 hours in in vitro test using a dialysis membrane.”,

Chitosan-ZnO nanocomposition hydrogels can contain insulin in their porous structure. They decreased the blood glucose level by 50-65% of the initial level through intranasal administration using rats. This hydrogel formulation containing insulin transformed from the solution to viscous hydrogel in the nasal cavity at body temperature (37°C). Such hydrogel state can reduce the mucociliary clearance rate and release slowly insulin there. Similarly, poly(DL-lactic acid-co-glycolic acid) /poly(ethylene glycol) /poly(DL-lactic acid-co-glycolic acid) (PLGA-PEG-PLGA) hydrogel (ICNPH) is the thermosensitive intelligent polymer, which is in a liquid state at room temperature (25°C) and in a sol-gel transition state at body temperature (37°C). The preparation of materials at room temperature is largely easy just by mixing liquid. Insulin-loaded chitosan nanoparticle/ICNPH (1 : 20 ratio) reduced retinal cell apoptosis greater compared to insulin alone in an in vivo assay using diabetic retinopathy model rats through a single subconjunctival injection. Insulin delivery system using ICNPH might be also effective to intranasal administration, because appropriate gelatinization on mucus layer of the epithelium delays mucociliary clearance. In addition, recently, albumin-based nanoparticles, as colloidal carrier potent to load electrostatically or covalently with cargos, have been used for nose-to-brain substance delivery. Human serum albumin is a biodegradable non-immunogenic protein (65–70 kDa). It is known that albumin transport pathways, mediated by gp60 or megalin/cubilin, function on the surface of endothelial and epithelial cells. Anti-inflammatory agents such as meloxicam (MEL), a selective COX-2 inhibitor, are effective to neuroinflammation in AD. MEL-loaded albumin nanoparticles showed higher cerebral concentration of MEL through intranasal administration in an in vivo assay using rats, compared to intravenous administration. Thus, this system based on albumin-based nanoparticles can be applied for intranasal insulin delivery.”,

and “In fact, insulin was delivered as cargo to the brain. However, in many cases insulin was released from nanoparticle within mucin layer. Successively, released insulin was transported into the epithelial cells across the membrane substantially as vector by itself. Therefore, it is suggested that insulin can be used for drug delivery of linked low molecular molecules.”

I added “From the findings about mucus layer permeation of insulin-loaded nanoparticles at the epithelium, insulin conjugates with bioactive substances particularly loaded on mucoadhesive nanoparticles and hydrogels would be distributed to the brain through nose-to-brain route greater than insulin conjugates with bioactive substances alone.”

Lines 510-528: The author proposes here the use of insulin conjugates to improve the brain delivery of some molecules with proven therapeutic effects. The author did not review the recent research progress of these strategies, but only presented this as a potential strategy. Thus, the author should reformulate the title of the article, as well as the abstract and the last lines of the introduction, which states that the purpose of this review is to discuss the use of insulin as a vector for intranasal delivery. I also suggest the author to reorganize the manuscript to highlight the nanodelivery strategies for insulin (section 2.6.3) adding other research works in this filed, and to further explore table 2. This table is very interesting and summarizes different studied delivery approaches for insulin (intranasal and other delivery routes). However, it is not well explored and discussed in the manuscript.

Response 9: I added “nanodelivery strategies for insulin” in title and abstract.

Intranasally administered unmodified insulin exhibited improvement of AD. Labelled-insulin derivatives were actually distributed to the brain through nose-to-brain route.

Moreover, I added other research works in section 2.6.3. ,

“Nanodelivery” as keyword, and “with or without nanoparticles using nano drug delivery systems” in conlusions.

I improved Table 2.

References list: Reference 71: the article title is wrong. Correct title is: “Monomeric Insulins and Their Experimental and Clinical Implications”

Response 10: I am sorry for this mistake. I corrected it.

That is all.